# Synthesis of CT Images Using CycleGANs: Enhancement of Anatomical Accuracy

**Dominik F. Bauer**                    Dominik.Bauer@medma.uni-heidelberg.de
**Alena-Kathrin Schnurr**        Alena-Kathrin.Schnurr@medma.uni-heidelberg.de
**Tom Russ**                                        Tom.Russ@medma.uni-heidelberg.de
**Stephan Goerttler**                Stephan.Goerttler@medma.uni-heidelberg.de
**Lothar R. Schad**                          Lothar.Schad@medma.uni-heidelberg.de
**Frank G. Zoellner**                        Frank.Zoellner@medma.uni-heidelberg.de
**Khanlian Chung**                        Khanlian.Chung@medma.uni-heidelberg.de
*Computer Assisted Clinical Medicine, Medical Faculty Mannheim, Heidelberg University, Germany*

## Abstract

Deep learning in medical imaging is often limited by the availability of training data with sufficient quality. The synthesis of image data offers a solution to this data shortage. Here, we use the CycleGAN network architecture to synthesize axial CT slices based on anthropomorphic body phantoms. We investigate the influence of an identity loss and a gradient difference loss function on the image quality of the synthesized data. We evaluate the synthesized images with respect to anatomical accuracy and realistic CT noise properties. The additional loss functions improved the preservation of edges and anatomical structures compared to the original CycleGAN loss, without deteriorating the noise quality of the synthetic image.

**Keywords:** CT Synthesis, CycleGAN, Simulation-Based Deep Learning

## 1. Introduction

The success of deep learning algorithms relies on the availability of training data with accurate annotations. However, due to data privacy regulations and the radiation exposure associated with CT imaging, there often is a shortage of image data in the field of medical image processing. One approach to obtain a large amount of annotated training data is the generation of synthetic data for the training procedure. It was already demonstrated that it is possible to achieve state-of-the-art results by solely training on a synthetic data set (Shrivastava et al., 2017). In the case of training data for CT image analysis tasks, morphological information of the human body has to be generated. This process raises several requirements regarding anatomical accuracy and variability in the synthetic CTs. Moreover, generating the CT specific noise texture is essential, as noise magnitude and noise texture play an important role in signal detection and can impact the performance of neural networks (Huang et al., 2018).

In this work, we utilize a CycleGAN network to synthesize CT image data based on XCAT body phantoms (Segars et al., 2010). The XCAT phantom allows the incorporation of anatomical variability in combination with pixel-precise annotation masks. The CycleGAN approach recently demonstrated feasible results for the translation from the MR to the CT

domain (Wolterink et al., 2017). In our application we observed, that the networks trained with the original CycleGAN loss often replaced high-contrast structures like bones and air cavities with soft tissue (see Figure 1). To enhance the preservation of these high-contrast structures, we extended the CycleGAN with two loss functions: an identity loss and a gradient difference loss. We compared the networks trained with the different loss functions by evaluating the anatomical accuracy and the noise properties of the synthetic CT images.

## 2. Methods

CycleGANs learn the mapping between two domains $X$ and $Y$ given unpaired training samples $x \in X$ and $y \in Y$ (Zhu et al., 2017). The mapping functions $G : X \to Y$ and $F : Y \to X$ are called generators. Two discriminators $D_X$ and $D_Y$ aim to distinguish between real images and generated images. The cycle consistency loss $L_{\mathrm{cyc}}(G, F)$ enforces forward and backward consistency for the generators, i.e. $F(G(x)) \approx x$ and $G(F(y)) \approx y$. With a least square generative adversarial loss $L_{\mathrm{adv}}(G, F, D_X, D_Y)$, the generators are trained to generate images which cannot be distinguished from real images by the discriminator. We additionally introduce the identity loss $L_{\mathrm{identity}}(G, F) = (||(G(x) - x)||_1 + ||(F(y) - y)||_1)$ and the gradient difference loss $L_{\mathrm{gdl}}(G, x) = \sum_{i,j} ||x_{i,j} - x_{i-1,j}| - |G(x)_{i,j} - G(x)_{i-1,j}||^2 + ||x_{i,j} - x_{i,j-1}| - |G(x)_{i,j} - G(x)_{i,j-1}||^2$ (Mathieu et al., 2015). The total generator loss is a combination of the previously defined losses with different weights:

$$L_{\mathrm{Generator}}(G, F, D_X, D_Y) = L_{\mathrm{adv}}(G, F, D_X, D_Y) + \lambda_{\mathrm{cyc}} L_{\mathrm{cyc}}(G, F)$$
$$+ \lambda_{\mathrm{identity}} L_{\mathrm{identity}}(G, F) + \lambda_{\mathrm{gdl}}(L_{\mathrm{gdl}}(G, x) + L_{\mathrm{gdl}}(F, y)).$$

We trained the CycleGAN with 4 empirically chosen combinations of weights $\lambda$, which are given in Table 1. For the generators we used a Res-Net architecture with 9 residual blocks. The discriminators are 70 x 70 PatchGANs, which were trained with a least square generative adversarial loss function (Zhu et al., 2017). We trained the networks with 20 contrast enhanced abdominal CT images, acquired in-house, and 20 XCAT abdomen phantoms for 150 epochs. We preprocessed the CT images by removing the patient table using a body contour segmentation. The XCAT phantoms were simulated with varying anatomical parameters. For the evaluation 10 additional XCAT phantoms were used. We calculated the structural similarity index (SSIM, (Wang et al., 2004)), feature similarity index (FSIM, (Zhang et al., 2011)), edge preservation ratio (EPR, (Chen et al., 2016)), and the mean absolute error (MAE) between the synthesized images and the phantoms to assess the anatomical accuracy. By using the labels of the XCAT phantom, we can easily calculate the MAE for different body regions. To compare the spatial correlation of the generated noise, we introduce the noise power spectra (NPS) correlation coefficient (NCC), which specifies the correlation between the radially averaged 2D-NPS of the synthetic and the real CT images. For a comparison of noise magnitudes, we calculated the standard deviation of the CT-numbers in liver segmentations.

## 3. Results

The evaluation metrics for the different loss function configurations are summarized in Table 1. All configurations show a realistic noise behaviour, with an NCC close to 1 and a noise

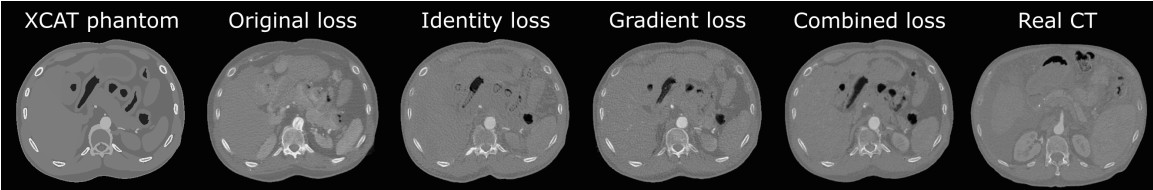

Figure 1: A slice of an XCAT phantom with the corresponding synthesized CT images from the 4 configurations. A real, unmatched CT image is given as a reference.

magnitude similar to the real CT images. The introduction of the identity and the gradient loss both lead to an increased FSIM, SSIM, EPR and a decreased MAE, which means that these loss functions help to preserve the essential anatomical information provided by the XCAT phantoms. Figure 1 shows a slice of an XCAT phantom and the corresponding synthesized CT images from each of the loss function configurations. It can be seen, that the additional losses lead to better preservation of the air cavities, especially for a combination of both losses. This observation is confirmed by the decrease of the MAE in air cavities by 54% for the combined loss.

Table 1: The image quality metrics for the evaluation of different training configurations. The noise magnitude of the real CT images is 44 HU.

|  | Original loss | Identity loss | Gradient loss | Combined loss |
| $\lambda_{\mathrm{cyc}}/\lambda_{\mathrm{identity}}/\lambda_{\mathrm{gdl}}$ | 10/0/0 | 10/5/0 | 10/0/10 | 10/5/5 |
| --- | --- | --- | --- | --- |
| FSIM | 0.60 | 0.66 | 0.65 | **0.71** |
| SSIM | 0.62 | 0.65 | 0.64 | **0.66** |
| EPR | 0.31 | 0.52 | 0.50 | **0.56** |
| MAE [HU] (Full body) | 102 | 73 | 84 | **65** |
| MAE [HU] (Bones) | 302 | 244 | 239 | **225** |
| MAE [HU] (Air cavities) | 261 | 148 | 125 | **121** |
| NCC | 0.98 | 0.98 | **0.99** | 0.99 |
| Noise magnitude [HU] | **43** | 29 | 41 | 36 |

## 4. Conclusion

Our results demonstrate that the introduction of the identity loss and gradient difference loss prevents anatomical inaccuracies in the synthetic CT images. A combination of both losses yielded a decrease of the MAE in bones by 25% and by 54% in the air cavities. However, the weighting parameters of the loss functions need to be chosen with care, since too large values cause the network to keep the XCAT images virtually unchanged. We observed realistic noise behaviour in all of the used configurations. In the future, the annotated synthetic CT images can be used to train neural networks, such as segmentation or registration networks.

## Acknowledgments

This research project is part of the Research Campus M$^2$OLIE and funded by the German Federal Ministry of Education and Research (BMBF) within the Framework "Forschungscampus - Public-Private Partnership for Innovation" under the funding code 13GW0388A.

We gratefully acknowledge the support of NVIDIA Corporation with the donation of the NVIDIA Titan Xp GPU used for this research.

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
