# OpenReview forum: "Synthesis of CT Images Using CycleGANs: Enhancement of Anatomical Accuracy"
_MIDL.io/2019/Conference/Abstract — MIDL Abstract 2019_

### Official Review · AnonReviewer2 · 2019-04-27
**Nice work using CycleGAN for CT image synthesis**

**Rating:** 3
**Confidence:** 3

**Review:**

Nice work using CycleGAN for CT image synthesis. Currently, however, only a single image is used for experiments. Also, it seems a challenge to objectively evaluate the performance, as no ground truth CT image is available for the XCAT phantom. It would be worthwhile to include more data and design a more appropriate evaluation metric in future work.

---

### Official Review · AnonReviewer1 · 2019-05-01
**loss function the key player !**

**Rating:** 3
**Confidence:** 2

**Review:**

This paper is about synthesizing CT images using cycleGAN, like many other similar works in the literature. However, the focus of this paper is different than that of existing studies: here the authorsa focus on the influence of an identity loss and a gradient difference loss function on the image quality of the synthesized data. Authors showed that the additional loss functions can improve the preservation of edges and anatomical structures compared to the original CycleGAN loss.

Experimental results, ablation study (limited), and comparisons (limited) are convincing, and the paper has a point in anatomical accuracy and realistic noise situations, which are necessary to our field of medical imaging. It seems that authors can make additional steps to introduce more loss functions that are specific to even tissues level and making it more unique to CT synthesis.

A small step to image synthesis, but a valid one, I believe.

---

### Decision · Program_Chairs · 2019-05-06
**Acceptance Decision**

Accept